# A Plant-Based Meal Increases Gastrointestinal Hormones and Satiety More Than an Energy- and Macronutrient-Matched Processed-Meat Meal in T2D, Obese, and Healthy Men: A Three-Group Randomized Crossover Study

**DOI:** 10.3390/nu11010157

**Published:** 2019-01-12

**Authors:** Marta Klementova, Lenka Thieme, Martin Haluzik, Renata Pavlovicova, Martin Hill, Terezie Pelikanova, Hana Kahleova

**Affiliations:** 1Institute for Clinical and Experimental Medicine, 140 21 Prague, Czech Republic; mkli@ikem.cz (M.K.); belenka@volny.cz (L.T.); halm@ikem.cz (M.H.); renata.pavlovicova@ikem.cz (R.P.); tepe@medicon.cz (T.P.); 2Institute of Endocrinology, 113 94 Prague, Czech Republic; mhill@endo.cz; 3Physicians Committee for Responsible Medicine, Washington, DC 20016, USA

**Keywords:** gastrointestinal hormones, nutrition, plant-based, satiety, type 2 diabetes

## Abstract

Gastrointestinal hormones are involved in regulation of glucose metabolism and satiety. We tested the acute effect of meal composition on these hormones in three population groups. A randomized crossover design was used to examine the effects of two energy- and macronutrient-matched meals: a processed-meat and cheese (M-meal) and a vegan meal with tofu (V-meal) on gastrointestinal hormones, and satiety in men with type 2 diabetes (T2D, *n* = 20), obese men (O, *n* = 20), and healthy men (H, *n* = 20). Plasma concentrations of glucagon-like peptide -1 (GLP-1), amylin, and peptide YY (PYY) were determined at 0, 30, 60, 120 and 180 min. Visual analogue scale was used to assess satiety. We used repeated-measures Analysis of variance (ANOVA) for statistical analysis. Postprandial secretion of GLP-1 increased after the V-meal in T2D (by 30.5%; 95%CI 21.2 to 40.7%; *p* < 0.001) and H (by 15.8%; 95%CI 8.6 to 23.5%; *p* = 0.01). Postprandial plasma concentrations of amylin increased in in all groups after the V-meal: by 15.7% in T2D (95%CI 11.8 to 19.6%; *p* < 0.001); by 11.5% in O (95%CI 7.8 to 15.3%; *p* = 0.03); and by 13.8% in H (95%CI 8.4 to 19.5%; *p* < 0.001). An increase in postprandial values of PYY after the V-meal was significant only in H (by 18.9%; 95%CI 7.5 to 31.3%; *p* = 0.03). Satiety was greater in all participants after the V-meal: by 9% in T2D (95%CI 4.4 to 13.6%; *p* = 0.004); by 18.7% in O (95%CI 12.8 to 24.6%; *p* < 0.001); and by 25% in H (95%CI 18.2 to 31.7%; *p* < 0.001). Our results indicate there is an increase in gut hormones and satiety, following consumption of a single plant-based meal with tofu when compared with an energy- and macronutrient-matched processed-meat meat and cheese meal, in healthy, obese and diabetic men.

## 1. Introduction

Obesity substantially increases the risk of type 2 diabetes, cardiovascular disease, and certain types of cancer [1]. Lifestyle change, including improved dietary choices, represents a primary prevention tool [2,3]. The influence of diet in the development of insulin resistance, prediabetes, and type 2 diabetes was investigated intensively [4,5,6]. Gastrointestinal hormones are involved in regulation of glucose metabolism, energy homeostasis, satiety, and weight management [7]. Ingestion of food triggers secretion of incretin hormones from the gastrointestinal tract, namely glucagon-like peptide-1 (GLP-1) and gastric inhibitory peptide (GIP), which enhance insulin secretion and help maintain glucose homeostasis [8]. Furthermore, the satiety hormones GLP-1 [9], peptide YY (PYY) [10], pancreatic polypeptide (PP) [11] and amylin [12] regulate appetite and energy homeostasis. The release of these satiety hormones can depend on meal composition and differs between impaired and normoglycemia [13].

Prospective observational studies indicate that consumption of red meat is positively associated with incidence of type 2 diabetes [14,15]. This association is particularly strong for processed meat. People consuming any processed meats have a one third higher likelihood to develop diabetes compared with those who do not consume any [16]. Several studies demonstrated a harmful effect of saturated fatty acids from meat and other animal products on insulin resistance and glucose tolerance [4,17] as well as other health issues such as cardiovascular disease [18]. In contrast, people following plant-based diets have their risk of diabetes cut in half, compared with non-vegetarians [19]. Several randomized clinical trials in participants with type 2 diabetes demonstrated a greater increase in insulin sensitivity and improvement in glycemic control with a plant-based diet when compared with a conventional diet [20,21].

We investigated postprandial metabolism in response to two meals matched for energy and macronutrient content: a processed-meat and cheese burger and a plant-based burger with tofu. We measured the physiological response to these meals in men with type 2 diabetes, body mass index (BMI) and age-matched obese normoglycemic men, and healthy age-matched controls. Our hypothesis is that a plant-based meal produces higher levels of gastrointestinal hormones and increased satiety in men with type 2 diabetes or obesity while having a negligible effect on healthy men. Our results will provide insight into the pathophysiologic mechanisms of the development of type 2 diabetes and will be used to inform the evidence base for dietary guidelines for men with type 2 diabetes.

## 2. Materials and Methods

### 2.1. Trial Design

We used a design of a randomized crossover study and enrolled 60 men; 20 men diagnosed with type 2 diabetes (T2D), 20 obese, BMI- and age-matched men (O), and 20 age-matched healthy controls (H) with two interventions. Screening was performed over the phone, contacting men from our hospital database who met the inclusion criteria. Eligible potential study participants came for an in-person meeting and were provided with details about the study participation. Recruitment was performed between June 2015 and June 2017. All men gave a written consent prior to enrollment in the study. This study was approved by the Ethics Committee of the Thomayer Hospital and Institute for Clinical and Experimental Medicine in Prague, Czech Republic on August 13, 2014. The protocol identification number is G14-08-42. The trial is prospectively registered with ClinicalTrials.gov (ID: NCT02474147).

### 2.2. Study Participants

All study participants were Caucasian men with a Czech nationality. Men with type 2 diabetes were between 30–65 years old. Their body mass index (BMI) was between 25–45 kg/m^2^ and they were treated by lifestyle alone or with oral hypoglycemic agents (metformin and/or sulfonylureas) for at least one year, who had an HbA1c from ≥42 to ≤105 mmol / mol, (≥6.0 to ≤11.8%) and at least three hallmarks of the metabolic syndrome. Our obese men were BMI- and age-matched to men with type 2 diabetes. Healthy men were age-matched controls within a healthy BMI range (BMI between 19–25 kg/m^2^) and with normal glucose tolerance. Exclusion criteria were thyroid, liver or kidney disease, drug or alcohol abuse, unstable drug therapy or a significant weight loss of more than 5% of body weight in the last three months.

### 2.3. Randomization and Masking

All men attended 2 study mornings. Before their first test morning, they were randomly assigned to an order of the test meals by the study physician (MK). Order of intervention was determined with a random sequence generator. The randomization protocol could not be accessed beforehand. The interventions were unmasked. MK was not involved in data analysis. Outcome assessors were blinded to the interventions.

### 2.4. Interventions

All men fasted for a minimum of 10 to 12 hours overnight. Men with type 2 diabetes were instructed to skip their diabetes medication the evening and the morning before the assessments. The meal consisted of either a processed meat and cheese burger (M-meal), or a plant-based tofu burger (V-meal). The staff delivered the meals fresh from the vendor. Tap water was allowed ad libitum. Plasma concentrations of gastrointestinal hormones, and self-reported satiety on a visual analogue scales were measured. Measurements were taken at baseline (time 0) and then 30, 60, 120 and 180 min after both meals.

### 2.5. Measurements

Anthropometric measures and blood pressure: A stadiometer was used to measure height and a periodically calibrated scale accurate to 0.1 kg was used to measure body weight. Waist circumference was measured with a tape measure placed at the midpoint between the lowest rib and the upper part of the iliac bone. Resting blood pressure was measured in a seated position after 5 min of resting, using a digital M6 Comfort monitor (Omron, Kyoto, Japan). Out of three measurements, the first measurement was disregarded, and a mean value was calculated for the remaining two measurements.

Gastrointestinal and appetite hormones. The concentrations of glucagon-like peptide-1 (GLP-1), amylin, and peptide YY (PYY) were measured using a Milliplex MAP Human Metabolic Hormone Magnetic Bead Panel (HMHEMAG-34K) (Millipore, Billerica, MA, USA) and a Luminex 100 IS analyzer (Luminex Corporation, Austin, TX, USA).

### 2.6. Statistical Analysis

Sample size was estimated based on a power analysis with an alpha of 0.05 and a power of 0.80 to detect between intervention differences in postprandial GLP-1 secretion. This required 14 participants in each group to complete both interventions. Intention to treat analysis was performed, using repeated-measures Analysis of variance (ANOVA). We included inter-individual factors (T2D vs. obese vs. healthy controls), intra-individual factors (time during the meal test) and interaction between factors (divergence degree between time profiles of each group) in the model. Spearman’s correlations were calculated to test the relationship between postprandial changes in gastrointestinal hormones. They were calculated for the fasting plus for changes (30–0, 60–30, 120–60, and 180–120 minutes after ingestion of standard meal): for each period separately, and then for all 5 values combined. Analyses were undertaken using PASS 2005 statistical software (Number Cruncher Statistical Systems, Kaysville, UT, USA), with the statistician blinded to the analyses. All results are presented as means with 95% confidence intervals (CI).

## 3. Results

The flow diagram is shown in Figure 1. Participant characteristics are shown Table 1. The meal composition is shown in Table 2.

### Gastrointestinal Hormones and Satiety

Postprandial secretion of GLP-1 increased after the V-meal in T2D (by 30.5%; 95%CI 21.2 to 40.7%; *p* < 0.001) and H (by 15.8%; 95%CI 8.6 to 23.5%; *p* = 0.01; Figure 2A). Postprandial plasma concentrations of amylin increased in all groups after the V-meal: by 15.7% in T2D (95%CI 11.8 to 19.6%; *p* < 0.001); by 11.5% in O (95%CI 7.8 to 15.3%; *p* = 0.03); and by 13.8% in H (95%CI 8.4 to 19.5%; *p* < 0.001; Figure 2B). An increase in postprandial values of PYY after the V-meal was significant only in H (by 18.9%; 95%CI 7.5 to 31.3%; *p* = 0.03; Figure 2C). Satiety was greater in all participants after the V-meal: by 9.0% in T2D (95%CI 4.4 to 13.6%; *p* = 0.004); by 18.7% in O (95%CI 12.8 to 24.6%; *p* < 0.001); and by 25.0% in H (95%CI 18.2 to 31.7%; *p* < 0.001; Figure 2D). A positive relationship was observed between Δ PYY and GLP-1 (*r* = 0.511, *p* < 0.001; Figure 3).

## 4. Discussion

Our study identified differences in markers of postprandial metabolism following the ingestion of a single plant-based meal when compared with a processed-meat meal that was matched for energy and macronutrient composition, particularly an increase in postprandial concentrations of GLP-1, amylin, and PYY. The differences were most noticeable in men with dysregulated glucose metabolism, those diagnosed with type 2 diabetes. Perhaps surprisingly, greater satiety was reported by all men following the V-meal. Contrary to our hypothesis, the difference between the meals was noticeable also in healthy volunteers. Our results provide comment on the interplay between components of gastrointestinal signalling during digestion in three separate participant groups.

We observed higher postprandial secretion of GLP-1 in obese men compared with healthy men. The highest concentrations of GLP-1 as a result of GLP-1 resistance were observed in men with T2D. GLP-1 was primarily associated, together with GIP, with the incretin effect, i.e. the increase in insulin secretion after meal ingestion in response to release of gut hormones [22]. The incretin effect is diminished in people with T2D [23,24] due to decreased beta-cell sensitivity [25]. Several hypotheses were formulated to explain loss of beta-cell sensitivity. Widely accepted concepts include hyperglycaemia- and hyperlipidaemia-associated receptor desensitization [26]. A few studies indicated that a high consumption of saturated fat increases the risk of impaired glucose tolerance [4,17] due to decreased beta-cell sensitivity and function. Preserving the capacity of beta-cells to secrete insulin in line with changing demand is a key goal in diabetes management [27]. Insulin secretion and beta-cell function may be enhanced, using different approaches that reduce body weight and fat (such as diet and exercise, bariatric surgery, or GLP-1 agonists) or favorably influence fat distribution (such as thiazolidinediones) [27,28]. As medications and bariatric surgery come with cost and potential side effects, lifestyle interventions should be the first-choice treatment. It was demonstrated that a 16-week plant-based diet intervention improves insulin resistance and beta-cell function in overweight subjects, addressing both core pathophysiologic mechanisms involved in diabetes at the same time [29]. Our results are in line with these hypotheses and further demonstrate that a plant-based meal may increase the postprandial concentrations of GLP-1, particularly in men with T2D.

The highest postprandial concentrations of PYY were observed in men with diabetes, being higher than in obese men and twice higher than in healthy men. PYY is synthesized and co-secreted together with GLP-1 [30]. PYY is a satiety hormone, because it decreases food intake and appetite after intravenous. administration [10] and the anorectic effects of PYY were pronounced even among obese and diabetic patients [31]. Furthermore, exaggerated secretion of PYY, together with GLP-1, plays a key role in metabolic improvements after bariatric surgery [32]. We found a positive relationship between the postprandial changes in PYY concentrations and changes in satiety in all men. Increased PYY secretion indicates an improved regulation of satiety and weight management in insulin-resistant individuals [33]. Its secretion is mostly dependent on the presence of free fatty acids in the small intestine [34]. Therefore, the rise in the PYY concentrations reflects the contact of lipids and other nutrients with the small intestine [35] and it is plausible to hypothesize that high concentrations of PYY in obese and diabetic men may simply reflect their baseline high-fat diet.

Postprandial amylin concentrations were higher in all men after the V-meal. The highest concentrations were observed among obese men, and the lowest in healthy ones. Amylin plays an important role in glucose metabolism and energy homeostasis [36] and is among satiety signals [37]. It is secreted in response to food and reduces eating by promoting satiation, together with GLP-1 and PYY. Interestingly, amylin analogues have been tested as an innovative treatment option of diabetes and obesity [38]. This further underlies the importance of postprandial increase in amylin concentrations, particularly in obese and diabetic men.

All men reported increased satiety after the V-meal. This might be partly explained by the fibre content of plant-foods, although most acute studies of meals differing in fibre consumption did not demonstrate enhanced satiety [39]. Since enhancing satiety is one of the major challenges in the dietary treatment of obesity and T2D, plant-based meals may be an effective strategy in solving this problem.

The main components responsible for the beneficial effects of a plant-based meal on satiety and the secretion of gastrointestinal hormones are fibre and bioactive compounds, such as polyphenols. Dietary fibre may increase PYY secretion and satiety in healthy people [40]. Additionally, resistant starch was shown to increase PYY concentrations in overweight individuals [41]. Furthermore, the available evidence suggests that polyphenols increase GLP-1 secretion, increase its half-life by inhibiting dipeptidyl peptidase-4, and stimulate β-cells to secrete insulin [42]. The effect of additional bioactive compounds of plant foods on satiety and secretion of gastrointestinal hormones is an emerging area of research.

The main strength of this study is our comprehensive measurement of the postprandial state. We identified differences in the signalling of mechanisms behind the health improvements associated with plant-based diets. We recorded markers of both signal and response to better consider the interplay of digestion and metabolism. We performed the same tests in men with type 2 diabetes, BMI and age-matched men, and age-matched healthy men so to consider differences in postprandial metabolism by state. Finally, our meals were commonly consumed meals served in quantities typically ingested, making the study results highly applicable and practical. This study also has several limitations. This is an acute meal test, with limited application to habitual diets and lifetime dietary patterns. However, when comparing these single meal responses, we identified metabolism differences that suggest longer-term studies would be beneficial to see if plant-based diets trigger additive or synergistic responses that improve markers of diabetes progression or prevent its incidence. Differences in the saturated fat and dietary fibre content of meals may have influenced our results. However, these differences are key when comparing plant-based diets with diets containing meat, so while this difference is not controlled for in this study, it does increase the generalisability of our results. In this study, the T2D and obese men were BMI-matched, while healthy controls had a significantly lower BMI. This difference in BMI may influence some of the responses we recorded.

## 5. Conclusions

Our findings indicate that plant-based meals with tofu may be an effective tool to increase postprandial secretion of gastrointestinal hormones, as well as promote satiety, compared to processed meat and cheese, in healthy, obese, and diabetic men. These positive properties may have practical implications for the prevention of type 2 diabetes.

## Figures and Tables

**Figure 1 nutrients-11-00157-f001:**
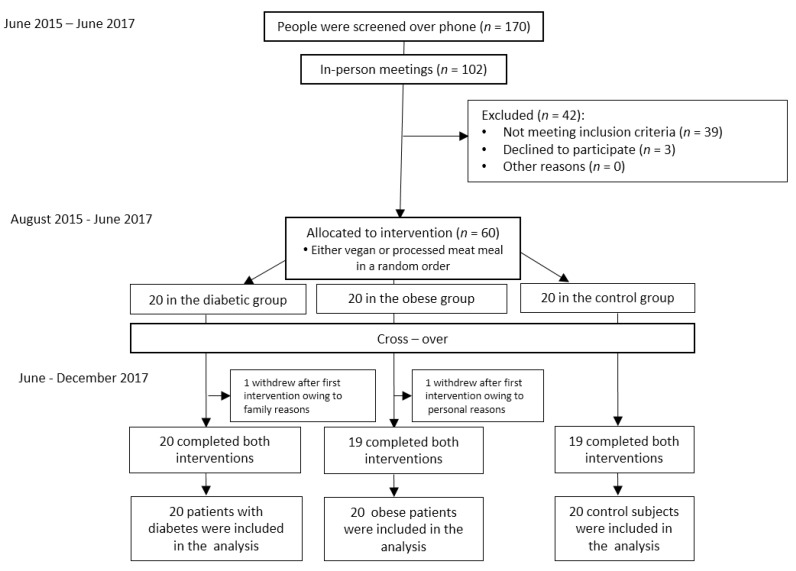
Enrollment of the participants and completion of the study.

**Figure 2 nutrients-11-00157-f002:**
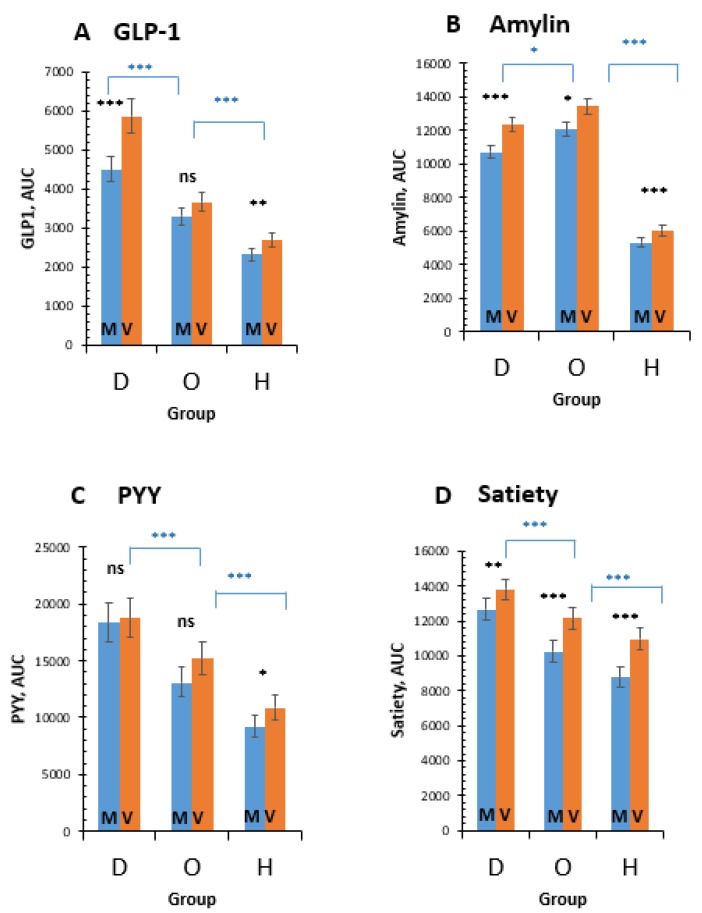
Postprandial changes in plasma concentrations of gastrointestinal hormones and in satiety in patients with diabetes (D), obese subjects (O), and healthy controls (H) after the ingestion of the processed-meat meat M-meal (M, blue) and the vegan V-meal (V, orange). Data are expressed as mean with 95% confidence intervals, using a repeated-measures Analysis of variance (ANOVA). * *p* < 0.05; ** *p* < 0.01; *** *p* < 0.001. (**A**): **GLP-1:** Meal: F = 19.3, *p* < 0.001; Group: F = 115.9, *p* < 0.001; Meal × Group: F = 1.2, *p* = 0.309; Subj (Group): F = 16.3, *p* < 0.001. (**B**): **Amylin:** Meal: F = 33, *p* < 0.001; Group: F = 469.1, *p* < 0.001; Meal × Group: F = 0.7, *p* = 0.506; Subj (Group): F = 21.3, *p* < 0.001. (**C**): **PYY:** Meal: F = 4, *p* = 0.05; Group: F = 40.7, *p* < 0.001; Meal × Group: F = 0.6, *p* = 0.532; Subj (Group): F = 17, *p* < 0.001. (**D**): **Satiety:** Meal: F = 25.6, *p* < 0.001; Group: F = 33, *p* < 0.001; Meal × Group: F = 0.8, *p* = 0.472; Subj (Group): F = 5.6, *p* < 0.001.

**Figure 3 nutrients-11-00157-f003:**
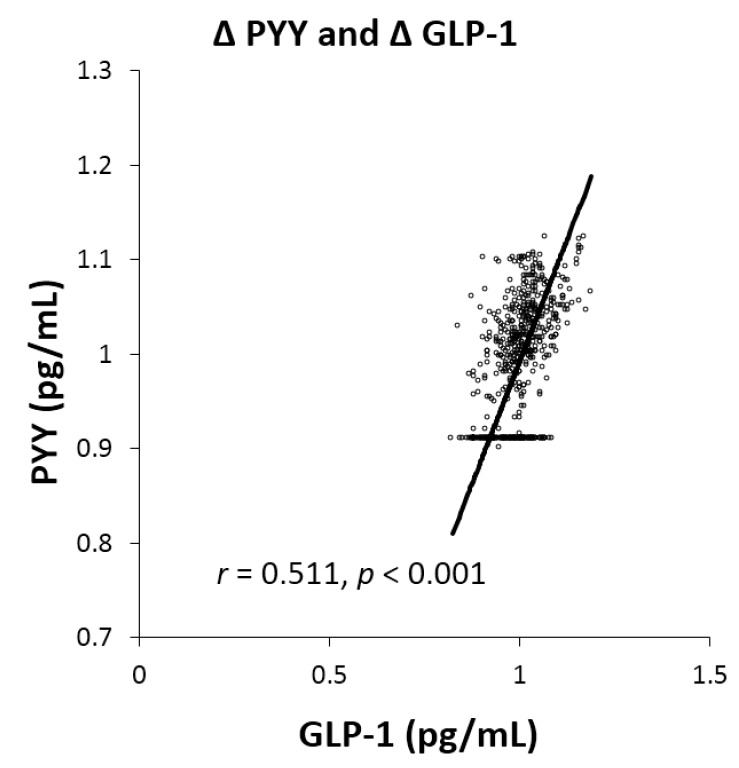
Correlation of postprandial concentrations of Δ GLP-1 and Δ PYY after both meals in all men. Spearman’s correlation was calculated for the relationship between changes in concentrations of investigated parameters. *r* = 0.511, *p* < 0.001.

**Table 1 nutrients-11-00157-t001:** Characteristics of the study population. Data are means ± standard deviation (SD).

Characteristic	Patients with T2D (*n* = 20)	Obese (*n* = 20)	Healthy Subjects (*n* = 20)
Age—years	47.8 ± 8.2	43 ± 7.0	42.7 ± 7.1
Weight—kg	108.2 ± 11.9	103.4 ± 13.3	77.4 ± 8.1
BMI—kg × m^−2^	34.5 ± 3.4	32.7 ± 3.9	23.8 ± 1.5
Waist—cm	106.9 ± 23.6	109 ± 8.5	85 ± 5.3
HbA1c (IFCC)—mmol/mol	48.5 ± 8.1	36.4 ± 3.0	36.1 ± 3.2
Fasting plasma glucose—mmol/L	7.2 ± 1.5	5.1 ± 0.3	5.1 ± 0.4
TGC—mmol/L	2.1 ± 1.1	2.2 ± 1.1	1.1 ± 0.6
LDL—mmol/L	2.6 ± 0.1	3.3 ± 0.7	2.8 ± 0.7
Blood pressure—mm Hg	144.4 ± 13.4/96.2 ± 8.8	134.8 ± 7.6/90 ± 6.8	124 ± 11.4/80.7 ± 5.6
Duration of diabetes—years	4.25 ± 3.25	-	-

**Table 2 nutrients-11-00157-t002:** Composition of the test meals.

Meal	M-meal	V-meal
Total weight (g)	200	200
Energy content (kCal)	513.6	514.9
Carbohydrates (g) (%)	55 (44.8%)	54.2 (44%)
Proteins (g) (%)	20.5 (16.7%)	19.9 (16.2%)
Lipids (g) (%)	22 (38.6%)	22.8 (39.8%)
Saturated fatty acids (g)	8.6	2.2
Fiber (g)	2.2	7.8

The postprandial state was measured after intake of a standard breakfast—one of two energy—(514 kcal) and macronutrient-matched meals (45% carbohydrates, 16% protein, and 39% lipids) in a random order: either a processed-meat burger meal (M-meal; cooked-pork seasoned meat in a wheat bun, tomato, cheddar-type cheese, lettuce, spicy sauce) together with 300 mL Café Latte with 21 g sugar, or a plant-based meal (V-meal; tofu burger with spices, ketchup, mustard, tomato, lettuce and cucumber in a wheat bun) together with 300 mL of unsweetened green tea.

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
