# Peer review of "A Plant-Based Meal Increases Gastrointestinal Hormones and Satiety More Than an Energy- and Macronutrient-Matched Processed-Meat Meal in T2D, Obese, and Healthy Men: A Three-Group Randomized Crossover Study"

_nutrients, 2019, doi:10.3390/nu11010157_

Reviewer 1 Report

Klementova et al have carried out a randomised controlled trial in people with T2DM, obesity or in healthy controls. The intervention and control meals were very tightly controlled in terms of macronutrients, and the study was very well designed.

Major points:

This study is very well designed. The authors should, however, state whether outcome assessors in the postprandial phase of the trial were blinded to the meal the participants were taking.

The figures (including the supplemental figure) were of very low resolution and this must be corrected for publication. It would also be great to have a visual representation of the statistically significant differences between the different groups (diabetes, obese, healthy), as this is a major and very interesting finding of the study.

The discussion was interesting, but would be greatly improved by the inclusion of more information on the potential components of the plant-based meal that could be responsible for the various effects. Is it plant-based meals per se, or simply tofu, that causes the effect? What can we conclude more generally from this study apart from the benefits of a tofu burger over a processed meat burger?

I would like to see more discussion on the differences between the 3 states: healthy, obese, diabetes. This was well done for GLP-1 but less so for the other outcomes.

Minor points:

The authors should use past tense throughout the manuscript, and some minor English editing is required.

The authors should follow the CONSORT 2010 checklist and include information such as:

Who did the randomisation, who knew about those randomised? Was any blinding done? Outcome assessors? What happened to the allocation after the random sequence was generated?

Author Response

Thank you for your helpful comments on our manuscript and for giving us the opportunity to further improve the paper. We have modified the manuscript accordingly and highlighted our changes in the manuscript in yellow.

Reviewer 2 Report

The submitted paper presented the results of a study on the differential effects of a processed meat vs. plant-based burger on gut hormone response and self-reported satiety among men in three groups: T2D, obesity, and healthy controls.  The writing was clear and concise and the results were easy to understand.  The topic is important as it relates to dietary variables that can help persons with T2D.

I recommend the following improvements:

  In the abstract the authors call the processed meat burger a "standard meat meal."  I strongly recommend adding the word "processed" before "meat," due to the greater negative health effects of processed meat vs. non-processed meat that the authors themselves alluded to in the introduction.  It is important to modify any discussion or generalization of findings to "processed meat" not meat in general, because this study looked specifically at processed meat, not non-processed meat.

Similarly, in the "Conclusion" section I recommend stating that plant-based meals may be effective... "compared to processed meat". 

I would also recommend replacing "standard meal" with "processed meat meal" in the title for greater clarification.

It is crucial that the authors make it very clear that all participants were men.  For example, in the title I strongly recommend replacing the word "participants" with "men," as these findings are not generalizable to women.

Similarly, this is a key change that needs to be made to the conclusion. It must be made very, very clear that only men were included in this study, and thus the results can only be generalizable to men.

More information is needed regarding recruitment methods, informed consent, race/ethnicity/nationality of participants, etc.  A very useful checklist of information to include about participants can be found at this link by the American Psychological Association, under the headings: "Method --> Sampling Procedures."  (https://www.apastyle.org/jars/quant-table-1.pdf ).   Additional guidelines specific to experimental studies (in addition to the previous link) can be found here: ( https://www.apastyle.org/jars/quant-table-2.pdf  ).

Fig. 1, 2, and the supplemental figure are blurry and difficult to read.  Please use larger/clearer images so readers can read them more easily. 

I highly recommend including the ingredients and/or sources for both the processed meat and plant-based burgers (including the buns, any condiments, etc.) so that the study can be replicated.  These would be a great addition to the supplementary file.

The figure in the supplementary file (participant flow chart) is usually included in the main article text.  It has very important methods information.

Finally, the authors report no conflicts of interest in this manuscript.  However, in Kahleova et al.'s paper published in Nutrients earlier this year (reference number 25 in the current manuscript), the following conflict of interest was reported which is very relevant to the current manuscript:  “Hana Kahleova works as the Director of Clinical Research at the Physicians Committee for Responsible Medicine, a nonprofit organization encouraging the use of low-fat, plant-based diets and discouraging the use of animal-derived, fatty, and sugary foods. Barnard has received research funding from the National Institute of Diabetes and Digestive and Kidney Diseases (NIH), the National Science Foundation, and the Diabetes Action Research and Education Foundation. He serves without financial compensation as president of the Physicians Committee for Responsible Medicine and Barnard Medical Center, nonprofit organizations providing education, research, and medical care related to nutrition. He writes books and gives lectures related to nutrition and health, and has received royalties and honoraria from these sources.”  Please adjust the conflicts of interest section as relevant for all authors.

Author Response

(The authors gave the same response as above.)
